# Demystifying MPNNs: Message Passing as Merely Efficient Matrix Multiplication

## Abstract

While Graph Neural Networks (GNNs) have achieved remarkable success, their design largely relies on empirical intuition rather than theoretical understanding. In this paper, we present a comprehensive analysis of GNN behavior through three fundamental aspects: (1) we establish that $k$-**layer** Message Passing Neural Networks efficiently aggregate $k$-**hop** neighborhood information through iterative computation, (2) analyze how different loop structures influence neighborhood computation, and (3) examine behavior across structure-feature hybrid and structure-only tasks. For deeper GNNs, we demonstrate that gradient-related issues, rather than just oversmoothing, can significantly impact performance in sparse graphs. We also analyze how different normalization schemes affect model performance and how GNNs make predictions with uniform node features, providing a theoretical framework that bridges the gap between empirical success and theoretical understanding.

## 1. Introduction

Graph Neural Networks (GNNs) are considered to be powerful in learning on graph-structured data, particularly through their iterative neighbor aggregation mechanism.

Despite their widespread adoption as feature extractors for graph data, fundamental questions about GNNs' representational capabilities remain open (Dehmamy et al., 2019). The design of new GNN architectures often relies on empirical intuition and heuristics rather than theoretical foundations (Xu et al., 2019).

While GNNs integrate both structural and feature information for predictions, our understanding of how these components interact and influence the final predictions remains limited. It is commonly assumed that a $k$-layer GNN effectively synthesizes both structural and feature information by aggregating data from progressively larger neighborhoods. However, our research reveals a more nuanced reality: when increasing from $k$ to $(k + 1)$ layers, the layer-wise iterative aggregation process effectively substitutes information from $k$-hop neighbors with that of $(k + 1)$-hop neighbors, rather than building a cumulative representation as previously thought. This is because graph loops lead to the coexistence of multi-hop neighbors in $k$-hop neighbors.

This paper demystifies Message Passing Neural Networks (MPNNs) by revealing their fundamental nature: the message passing process is, at its core, a memory-efficient implementation of matrix multiplication operations. Through this lens, we demonstrate three key insights:

(1) In Section 2, we establish that a $k$-layer MPNN transforms node representations by iteratively aggregating information from $k$-hop neighborhoods.

More precisely, we prove an approximation equivalence: a $k$-layer MPNN operating with adjacency matrix $A$ is approximately equivalent to a single-layer MPNN operating with adjacency matrix $A^k$.

This result not only provides a formal characterization of how message passing depth relates to neighborhood influence in GNNs, but also reveals a computational advantage: while direct computation of $A^k$ requires storing the full power matrix and can exceed memory constraints for large graphs, the iterative message passing in GNNs achieves equivalent neighborhood aggregation through memory-efficient layer-wise operations.

(2) In Section 3, we analyze how different types of graph loops affect $k$-hop neighborhood computation, as loops create additional paths between nodes and thus increase the density of $k$-hop neighborhoods.

(3) Finally, in Section 4, we examine MPNN behavior across structure-feature hybrid tasks and structure-only tasks, revealing their underlying similarity: structure-only tasks are essentially structure-feature hybrid tasks where node degrees serve as the node features.

---

[1]Anonymous Institution, Anonymous City, Anonymous Region, Anonymous Country. Correspondence to: Anonymous Author <anon.email@domain.com>.

Preliminary work. Under review by the International Conference on Machine Learning (ICML). Do not distribute.

We challenge the conventional wisdom about deeper GNNs' performance degradation: contrary to the common over-smoothing (Rusch et al., 2023) explanation, we experimentally demonstrate that gradient-related issues can be the primary cause for sparse graphs. In addition, we explain how GNNs predict with uniform features and how different normalization schemes fundamentally influence their performance.

In summary, our work provides a theoretical foundation for understanding GNN behavior through three key aspects: the relationship between network depth and neighborhood aggregation, the impact of graph loop structures, and the role of gradients in deep architectures, normalization influence. These theoretical insights not only bridge the gap between empirical success and mathematical understanding but also provide practical guidance for GNN architecture design and deployment across various applications.

The code for the experiments conducted in this paper is available at https://anonymous.4open.science/status/demystify-B30E.

**Notation and definitions**   A graph $G = (A, X)$ is a set of $N$ nodes connected via a set of edges. The adjacency matrix of a graph $A$ encodes graph topology, where each element $A_{ij}$ represents an edge from node $i$ to node $j$. In this paper, edges are directed, the undirected graph is considered to be a special case of directed graph where all edges have their reversed edges in the graph. Each node $i$ is assigned a feature vector $\mathbf{x}_i \in \mathbb{R}^d$, and all the feature vectors are stacked to a feature matrix $\mathbf{X} \in \mathbb{R}^{n \times d}$, where $n$ is the number of nodes in $G$. The set of neighbors of node $i$ is denoted by $\mathcal{N}(i)$.

We use $AB$ or $A \cdot B$ to denote the matrix product of matrices $A$ and $B$. All multiplications and exponentiations are matrix products, unless explicitly stated. Lower indices $A_{ij}$ denote $i, j$th elements of $A$, and $A_i$ means the $i$th row. $A^p$ denotes the $p$th matrix power of $A$.

## 2. $k$-layer GNNs

### 2.1. $k$-order features

**Definition 2.1.** The $k$-**hop neighbor** of a node $v$ in a graph $G = (V, E)$ is any node $u \in V$ such that there is a directed path of $k$ consecutive edges from node $u$ to node $v$.

**Definition 2.2.** A $k$th order node feature, defined as $A^k X$, represents the result of multiplying the adjacency matrix $A$ with itself p times and then multiplying with the node feature matrix $X$. Particularly, 0th order node feature is the original node feature.

**Lemma 2.3.** *For a graph $G = (V, E)$ with adjacency matrix $A$ and node feature matrix $X$, the features aggregated*

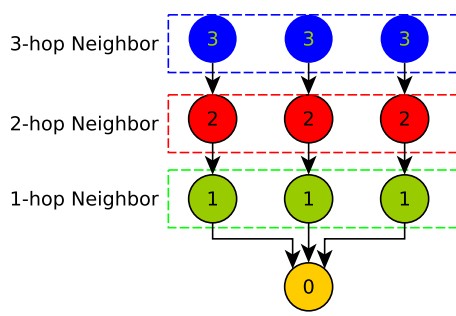

**Figure 1.** A $k$-layer GCN without adding selfloop will only gather information from $k$-hop neibors.

*from p-hop neighbors of each node are equivalent to the kth order node feature $A^k X$.*

*Remark* 2.4.  $A^k X W$ is a linear transformation of $k$-hop neighbor features $A^k X$ using weight matrix $W$.

**Lemma 2.5.** *In the $k$-th power of the adjacency matrix $A^k$, a non-zero element $A_{ij}^k > 0$ indicates that there exists at least one directed path of length exactly $k$ from node $i$ to node $j$. Furthermore, the value of $A_{ij}^k$ represents the total number of such paths.*

The proof is provided in Appendix A.1.

*Remark* 2.6. The $k$th order node feature gathers information from nodes which are exactly $k$-hop away from the center node, as illustrated in Fig. 1.

### 2.2. Node representation of $k$-layer GNNs

**Lemma 2.7.** *For all natural numbers $k$, the output of a $k$-layer GCN without self-loops can be expressed as:*

$$H^{(k)} = \sigma\big((W \odot A)^k X W^{(k)}\big) \tag{1}$$

The proof is provided in Appendix A.3.

**Lemma 2.8.** *For all natural numbers $k$, the output of a $k$-layer GCN with self-loops can be expressed as:*

$$H^{(k)} = \sigma\big((W \odot (A + I))^k X W^{(k)}\big) \tag{2}$$

The proof is provided in Appendix A.2.

$(A + I)^k$ can indeed be decomposed into a linear combination of powers of $A$, as described by the binomial theorem:

$$(A + I)^k = \sum_{k=0}^{k} \binom{n}{k} A^k$$

Where $\binom{n}{k}$ is the binomial coefficient.

Therefore, the final representation $H^{(k)}$ is a linear combination of the feature transformations derived from paths of

lengths ranging from 0-th to $k$-th order, capturing information aggregated over different scales in the graph.

**Lemma 2.9.** *For all natural numbers $k$, the output of a $k$-layer GraphSAGE can be expressed as:*

$$H^{(k)} = \sigma\big((W \odot A)^k XW_n^{(k)} + ... + (W \odot A)XW_1^n + XW_0^{(k)}\big)$$
(3)

The proof is provided in Appendix A.4.

The final representation of a $k$-layer GraphSAGE, similar to a GCN with self-loops, is derived from a combination of linear transformations applied to graph features aggregated from 0-th to $k$-th order neighborhoods.

### 2.3. Summary of $k$-layer GNNs

In summary, for a $k$-layer GNN, both GCN with self-loops and GraphSAGE integrate information from all neighborhood orders up to $k$. In contrast, a GCN without self-loops incorporates information solely from the $k$-th order neighborhood, as lower-order features are excluded in the absence of self-loops.

The approximation capabilities of graph neural networks (GNNs) reveal that a $k$-layer GNN with an adjacency matrix $A$ has the same approximation power as a 1-layer GCN with the adjacency matrix $A^k$. This observation demystifies the iterative aggregation power of message-passing neural networks (MPNNs).

In essence, multiple iterations of aggregation are equivalent to performing high-order matrix multiplications.

However, adding self-loops (as in GCNs with self-loops) or concatenating self-node features (as in GraphSAGE) incorporates features of all orders. While this can enhance the expressiveness of the model, it may also lead to over-smoothing, ultimately limiting the depth of GNNs and their ability to capture meaningful representations in deeper architectures.

## 3. Loops

In Section 2, we discussed the influence of self-loops in GCNs. In this section, we will extend our discussion to consider all types of loops in graph neural networks and analyze their effects.

### 3.1. Self-loops

Sources of self-loops include:

1. **Original Graph**: In some networks, such as webpage networks, a node (e.g., a webpage) might naturally link to itself.

2. **GNN Model Design**: Many GNN models, such as

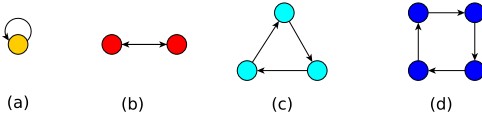

*Figure 2.* Types of loops in graphs: (a) self-loop, (b) loop with two nodes connected by an undirected edge, (c) and (d) are examples of n-node loops where n=3 and n=4 respectively.

GCN (Kipf & Welling, 2016) and DiG(ib) (Tong et al., 2020), explicitly add self-loops to improve performance, particularly on homophilic graphs.

As discussed in Sec. 2.8, a $k$-layer GCN with self-loops would gather information from neighbors within the range of 0-hop to $k$-hop neighbors. This fact was established via matrix multiplication. In this section, we will prove it geometrically.

**Lemma 3.1.** *When self-loops are added to a graph, the $k$-hop neighbors of any node are also its $(k+1)$-hop neighbors.*

The proof is provided in Appendix B.1. This path-based property can be expressed in terms of the adjacency matrix:

**Lemma 3.2.** *Let $G$ be a graph with self-loops. Then for any $k \geq 1$, any connection present in $A^k$ is also present in $A^{k+1}$.*

### 3.2. Two-node Loops

A directed graph where each pair of connected nodes has edges in both directions (making its adjacency matrix symmetric) can be viewed as an undirected graph. In other words, an undirected graph is equivalent to a directed graph where every edge is bidirected.

**Lemma 3.3.** *For an undirected graph, for any $k \geq 1$, the $k$-hop neighbors of any node are also its $(k+2)$-hop neighbors.*

The proof is provided in Appendix B.2. This property can be expressed in terms of the adjacency matrix:

**Lemma 3.4.** *Let $G$ be an undirected graph. Then for any $k \geq 1$, any connection present in $A^k$ is also present in $A^{k+2}$.*

### 3.3. Multi-node Loops

**Lemma 3.5.** *For a graph containing a loop of length $m$, let $v$ be any node in the graph. For any $k \geq 1$, if $u$ is a $k$-hop neighbor of $v$ where the $k$-hop path from $u$ to $v$ contains at least one node from the loop, then $u$ is also a $(k+m)$-hop neighbor of $v$.*

The proof is provided in Appendix B.3. This path-based

property can be naturally expressed in terms of the adjacency matrix of the graph:

**Lemma 3.6.** *Let $G$ be a graph containing a loop of length $m$. Then for any $k \geq 1$, any connection present in $A^k$ is also present in $A^{k+m}$.*

### 3.4. Longest path

#### 3.4.1. FOR DIRECTED GRAPH

**Lemma 3.7.** *For a directed graph with adjacency matrix $A$, if the graph contains no loops (cycles) and $h$ is the length of the longest simple path, then:*

$$A^m = 0 \text{ for all } m > h$$

The proof is provided in Appendix B.4.

#### 3.4.2. FOR UNDIRECTED GRAPH

**Lemma 3.8.** *Let $G$ be an undirected graph, and let $h$ be the length of the longest path in $G$. Then for any $m > h$, the connections present in $A^m$ are identical to those in $A^h$.*

The proof is provided in Appendix B.5.

### 3.5. Loops Influence

Different types of graph structures influence how connectivity patterns evolve as we take higher powers of the adjacency matrix. Self-loops allow paths to extend by single steps, while two-node loops (undirected edges) enable extension by pairs of steps. More generally, any $m$-node loop allows paths to extend by $m$ steps while preserving all existing connections.

For the maximal path length, undirected and directed graphs behave quite differently. In undirected graphs, paths can always extend beyond the spanning tree's longest path length while maintaining the same connectivity pattern. However, in directed acyclic graphs, no paths can exist beyond this length.

Table 1 summarizes these relationships, where $\overset{c}{=}$ denotes identical connectivity patterns in the adjacency matrices.

We provide a mathematical analysis of how different neighborhood hops coexist in $k$-layer GNNs, a phenomenon that can contribute to over-smoothing. Our analysis identifies three critical factors that drive this coexistence: (1) Adding self-loops; (2) Undirected edges; (3) Presence of loops in datasets.

**Empirical Validation** Common GNN preprocessing steps—adding self-loops and symmetrizing directed graphs through reverse edges—significantly increase the density of $k$-hop neighborhoods. Our experiments demonstrate that avoiding these modifications can prevent over-smoothing.

| Graph Type | Matrix Connectivity |
|---|---|
| Self-loops | $A^k \overset{c}{=} A^{k+1}$ |
| Two-node loops | $A^k \overset{c}{=} A^{k+2}$ |
| $n$-node loops | $A^k \overset{c}{=} A^{k+n}$ |
| Undirected | $A^m \overset{c}{=} A^h$ for $m > h$ |
| Directed acyclic | $A^m = 0$ for $m > h$ |

*Table 1.* Graph Types and Matrix Connectivity. Here $\overset{c}{=}$ denotes that two matrices have identical connectivity patterns (presence of non-zero entries), $h$ is the length of the longest path, and $m$ is any hop count value.

As shown in Fig. 3, a GCN model without self-loops or undirected conversion maintains performance even at 50 layers, whereas standard GNNs typically experience over-smoothing within 5 layers (Li et al., 2018). These results align with our theoretical analysis showing that self-loops and undirected edges induce neighborhood hop coexistence, a potential mechanism for over-smoothing.

More information about datasets and experiments are presented in Appendix C.1.

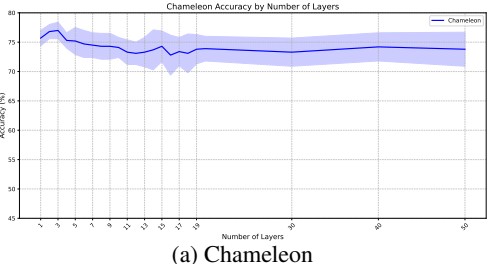

(a) Chameleon

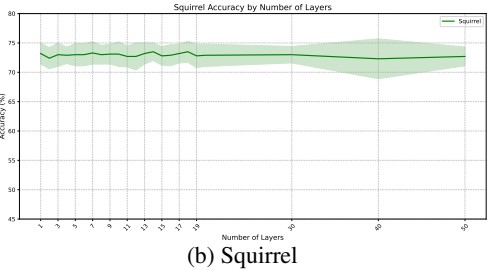

(b) Squirrel

*Figure 3.* Performance of Unidirectional GCN Without Self-loops on Chameleon and Squirrel Datasets: Model demonstrates stable accuracy up to 50 layers, with deeper architectures constrained by memory limitations. The solid line represents mean accuracy, while the shaded region indicates standard deviation across 10 data splits.

## 4. Structure-Feature Dichotomy in Node Classification

Graph Neural Networks (GNNs) combine node features and graph structure for predictions. However, recent work shows

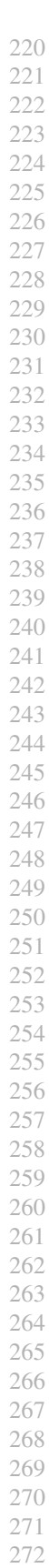

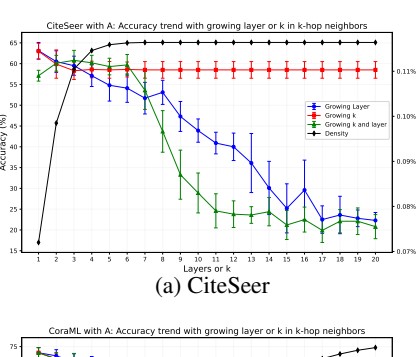

(a) CiteSeer

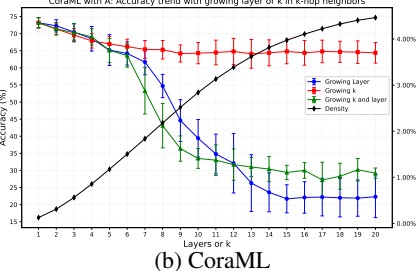

(b) CoraML

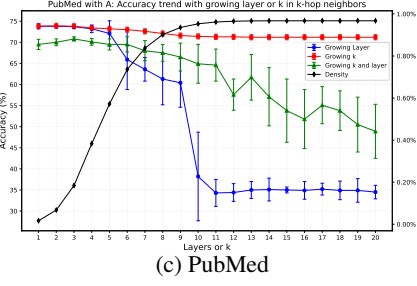

(c) PubMed

*Figure 4.* Comparison of different GCN architectures on three datasets: $k$-layer GCN (**blue**), 1-layer GCN with $k$-hop neighbors (**red**), and $k$-hop neighbors with 1-layer GCN and ($k$-1) linear layers (**green**). The black line shows the density of $k$-hop adjacency matrix.

structure-agnostic models like MLPs outperform GNNs on certain datasets (e.g., WebKB (Zheng et al., 2022)). Complementing this finding, we show that some node classification tasks perform equally well without node features. Based on this Structure-Feature Dichotomy, we categorize tasks into three types: feature-only, structure-only, and hybrid. We then analyze how GNNs make predictions for the latter two cases.

## 4.1. Structure-feature Hybrid Type

Citation networks like Cora, CiteSeer, and PubMed represent classic node classification tasks where research papers are categorized by their topics. While individual papers may not contain comprehensive field-specific content, aggregating features from neighboring nodes can enrich the representation of each paper's research domain, making GNNs particularly effective for this task.

Figure 4 shows consistent patterns across CiteSeer, CoraML,

and PubMed datasets comparing three approaches: (1) increasing GNN layers with first-order neighbors (blue line), (2) single-layer GNN with increasing $k$-hop neighbors (red line), and (3) $k$-hop neighbors with ($k$-1) additional linear layers (green line). The black line represents the density of the effective adjacency matrix—the percentage of non-zero elements after $k$-hop expansion. The low density values across all datasets indicate these are sparse directed networks.

The single-layer GNN with increasing $k$-hop neighbors maintains stable performance, while both the deep GNN and the hybrid approach show significant performance degradation with increasing depth. While both architectures access $k$-hop neighborhood information—through $A^k$ in single-layer GNN and k successive applications of $A$ in $k$-layer GNN—their empirical performance differs substantially despite theoretical equivalence in terms of Universal Approximation (Section 2.3).

The fact that performance remains stable when increasing the neighborhood size k in a single-layer architecture (red line) indicates minimal over-smoothing in this case. Thus, we hypothesized that gradient-related issues might be the primary cause of performance degradation in deeper networks.

To test this hypothesis, we designed approach (3) which combines $k$-hop neighborhood aggregation with ($k$-1) additional linear layers. This architecture shares parameter count with the deep GNN while using expanded neighborhoods like the single-layer approach. The deteriorating performance of this hybrid approach parallels that of the deep GNN, strongly suggesting that gradient-related issues, rather than over-smoothing, cause the performance drop in deep GNNs.

We further investigated Reverse Direction and Bidirectional Propagation, with experimental results on CiteSeer presented in Figure 5. For Reverse Direction Propagation, we observed performance trends similar to forward direction propagation. However, Bidirectional Propagation exhibited distinct behavior: increasing the neighborhood size k in single-layer architectures (red line) led to performance degradation, which can be attributed to over-smoothing since the connection density inversely correlates with k. The connection density (black line) and performance with increasing k (red line) both stabilized after k=17.

As shown in Figure 5b, $k$-layer GNNs with first-order neighbors (blue line) performed worse than single-layer models with equivalent $k$-hop neighbors (red line), likely due to the compound effects of over-smoothing and vanishing gradients.

Experimental results for the CoraML and PubMed datasets are detailed in Appendix D. While Reverse Direction Prop-

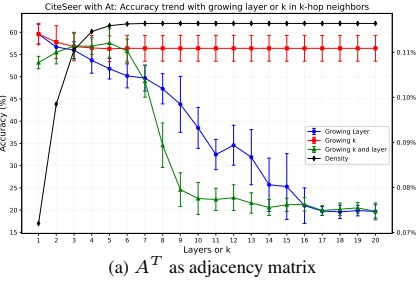

(a) $A^T$ as adjacency matrix

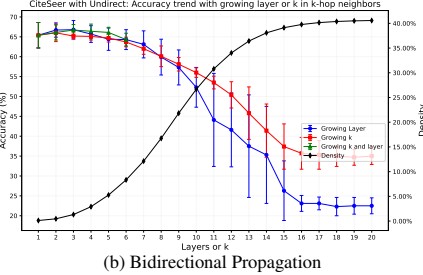

(b) Bidirectional Propagation

*Figure 5.* Comparison of different GCN architectures on CiteSeer dataset under different adjacency matrix formulations. (Top) Using transposed adjacency matrix $A^T$, which propagates information from cited papers to citing papers. (Bottom) Using undirected graph adjacency matrix $A + A^T$, which enables bidirectional information flow. In each subplot: k-layer GCN (blue), 1-layer GCN with k-hop neighbors (red), and k-hop neighbors with (k-1) linear layers (green). The black line indicates the density of the k-hop adjacency matrix.

agation exhibits performance trends similar to our main findings, Bidirectional Propagation demonstrates significant over-smoothing behavior. Specifically, in the bidirectional case, neighborhood density exceeds 80% within 8 hops, leading to degraded performance. Consequently, single-layer models utilizing equivalent $k$-hop neighborhoods underperform compared to $k$-layer GNNs that only aggregate first-order neighbors.

Thus, while our findings hold true for sparse networks, dense networks exhibit different behavior. The distinct performance patterns in dense networks suggest that over-smoothing may still play a significant role in their degradation.

These results enhance our understanding of GNN performance degradation, indicating that optimizing architectural design for effective gradient flow may be more crucial than addressing over-smoothing effects in sparse networks, while over-smoothing remains a key consideration for dense networks.

### 4.2. Structure-only Type

In this section, we will present three datasets which work well without node features, where all nodes have uniform features.

*Table 2.* Classification accuracy (%) of Dir-GNN (Rossi et al., 2024) with different feature configurations and normalization schemes on Chameleon, Squirrel and Telegram datasets. Feature configurations include: original node features from datasets (Origin Feature), constant features (No Feature, all set to 1), and node degree variants (in-degree, out-degree, or both). **Bold** values indicate learning failure with row normalization and no features. Underlined values show the worst performance among all configurations, except for the case of row normalization with no features, which can be attributed to numerical instability when normalization is absent.

| Chameleon | Feature | None | Row | Sym | Dir |
|---|---|---|---|---|---|
| MLP | | | 48.0±1.6 | | |
| Origin Feature | 2,325 | 79.1±1.4 | 80.0±1.5 | 79.4±1.6 | 79.8±1.4 |
| No Feature | 1 | 73.6±2.4 | **23.0±2.6** | 77.9±1.8 | 78.1±1.2 |
| In-degree | 1 | 75.2±2.0 | 77.0±2.0 | 78.0±2.5 | 78.1±1.8 |
| Out-degree | 1 | 73.6±2.0 | 77.8±1.5 | 78.0±2.3 | 77.6±2.1 |
| Both degrees | 2 | 75.5±1.8 | 77.6±1.5 | 77.9±2.2 | 77.4±1.3 |

| Squirrel | Feature | None | Row | Sym | Dir |
|---|---|---|---|---|---|
| MLP | | | 36.3±1.5 | | |
| Origin Feature | 2,089 | 74.3±2.3 | 75.1±1.6 | 76.3±1.9 | 76.1±2.0 |
| No Feature | 1 | 67.8±3.4 | **19.5±1.1** | 75.5±2.3 | 75.6±2.0 |
| In-degree | 1 | 64.9±5.8 | 73.0±3.0 | 75.7±1.7 | 75.3±1.6 |
| Out-degree | 1 | 63.7±4.9 | 72.2±3.8 | 75.5±1.6 | 75.1±1.8 |
| Both degrees | 2 | 67.9±3.9 | 73.1±3.1 | 76.1±1.4 | 75.4±1.8 |

| Telegram | Feature | None | Row | Sym | Dir |
|---|---|---|---|---|---|
| Origin Feature | 1 | 95.6±2.8 | 74.2±5.5 | 93.0±4.1 | 92.8±4.7 |
| No Feature | 1 | 95.4±4.0 | **38.0±0.0** | 93.0±4.7 | 93.0±3.0 |

As shown in Table 2, for Chameleon, Squirrel and Telegram datasets, Dir-GNN [1] (Rossi et al., 2024) predicts as well with no feature as with original features. We will give a brief summarization of existing normalizations.

### 4.2.1. NORMALIZATIONS

Graph normalization, which typically involves dot multiplication of the adjacency matrix to adjust edge weights, plays a crucial role in graph neural networks (GNNs). While various normalization schemes exist, their theoretical implications remain under-explored. We denote a general normalization function as $f(A)$.

**No Normalization**  The simplest approach is to use the raw adjacency matrix without any normalization (Li et al., 2017): $f_1(A) = A$. In this case, the node feature update rule becomes:

$$h_i^{(l+1)} = \sigma\left(\sum_{j \in \mathcal{N}(i)} h_j^{(l)} W^{(l)}\right)$$

The aggregation directly sums neighboring features, leading to larger feature magnitudes for higher degree nodes. With homogeneous features $h_i^{(0)} = 1$, node representations become proportional to degrees.

While this makes it suitable for degree-dependent tasks like traffic prediction and network flow classification, repeated aggregation of unnormalized features can cause numerical instability. The node representations may grow or vanish exponentially with network depth. This numerical instability explains the suboptimal performance of unnormalized adjacency matrices compared to normalized variants in Table 2. The exponential growth or decay of node representations across layers likely hindered the model's ability to learn effective graph representations, despite preserving the degree information.

**Row Normalization**  Row normalization (Hamilton et al., 2017) scales each row of the adjacency matrix by the inverse of node degree: $f_2(A) = D^{-1}A$. The node feature update rule becomes:

$$h_i^{(l+1)} = \sigma\left(\frac{\sum_{j \in \mathcal{N}(i)} h_j^{(l)} W^{(l)}}{d_i}\right)$$

For this formulation, the aggregated information represents the mean of neighboring features rather than their sum. Node degrees no longer directly influence feature magnitudes. With homogeneous features $h_i^{(0)} = 1$, all nodes get identical representations. This explains the poor traffic prediction in Table 2—degree information is lost.

---

[1]Specific parameter settings are shown in ScaleNet (Jiang et al., 2024) . For Telegram datasets, self-loops are added for better performance.

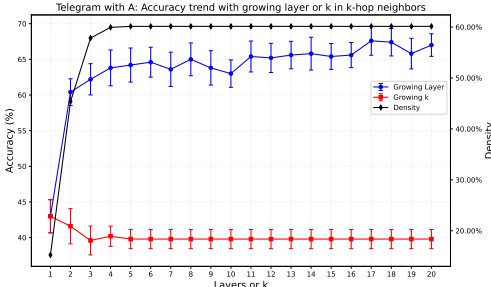

*Figure 6.* On Telegram dataset, layer growth achieves good accuracy while k growth performs poorly with no predictive power. Density shows the percentage of non-zero elements in the equivalent adjacency matrix.

**Symmetric Normalization**  Symmetric normalization (Kipf & Welling, 2016) applies: $f_3(A) = D^{-1/2}AD^{-1/2}$. The node feature update rule becomes:

$$h_i^{(l+1)} = \sigma\left(\sum_{j \in \mathcal{N}(i)} \frac{h_j^{(l)} W^{(l)}}{\sqrt{d_i d_j}}\right)$$

The neighbor's influence is determined by both degrees—if a neighbor's degree is much larger than the center node's, its feature weight becomes smaller than in row normalization.

**Directed Normalization**  For directed graphs, Rossi et al. (2024) proposes: $f_4(A) = D_{in}^{-1/2}AD_{out}^{-1/2}$. The node feature update rule becomes:

$$h_i^{(l+1)} = \sigma\left(\sum_{j \in \mathcal{N}(i)} \frac{h_j^{(l)} W^{(l)}}{\sqrt{d_i^{in} d_j^{out}}}\right)$$

This distinguishes between in-degree and out-degree for more accurate normalization in directed graphs.

In summary, row normalization of adjacency matrices, when applied with uniform node features, results in loss of structural information since all nodes become indistinguishable. While using the unnormalized adjacency matrix preserves both degree information and feature distinctions, it can lead to numerical instability since the eigenvalues of $f(A)$ may grow or diminish exponentially, rather than being bounded within [-1, 1]. This instability can affect the training of graph neural networks.

### 4.2.2. HOW GNN PREDICT FOR STRUCTURE-ONLY CLASSIFICATION

Fig. 6 demonstrates that increasing the number of GCN layers leads to better accuracy, while increasing the power $k$ of the adjacency matrix ($k$-hop neighborhoods) results in poor predictive performance on the Telegram dataset.

As shown in Table 2, models using uniform features (all set to 1) achieve comparable performance to those using original features, suggesting that the original node features contribute little to predictive power. We therefore focus our analysis on the simpler uniform feature case.

With uniform features set to 1, we observe two scenarios:

- Single-layer GCN with $k$-hop adjacency matrix ($A^k$): The node representations simply become counts of $k$-hop neighbors. The model loses predictive power beyond $k > 2$.

- $k$-layer GCN with standard adjacency matrix ($A$): After the first layer, node representations become equivalent to node degrees. In subsequent layers, representations capture the degrees of neighboring nodes. This effectively means a $(k+1)$-layer GNN without features is equivalent to a $k$-layer GNN using node degrees as the only feature.

This explains the patterns observed in Figure 6. The single-layer GNN performs poorly because it only considers individual node degrees without incorporating neighborhood information. From two layers onward, performance improves steadily as the effective adjacency matrix becomes denser, eventually stabilizing when additional layers no longer increase the density of connections.

### 4.3. Summary

In this section, we further analyze MPNN predictions through the structure-feature dichotomy. Node classification datasets can be divided into a structure-feature dichotomy, where some datasets perform better with MLPs. For datasets where GNNs are effective, predictions can be made with or without node features, determined by the task type.

**Structure-feature Hybrid Tasks** Content-based classification problems, exemplified by citation networks, require integration of both node features and neighborhood features propagated through structural connections. Our analysis reveals that the commonly observed performance degradation in deeper layers, traditionally attributed to over-smoothing, may instead stem from gradient-related challenges, especially for sparse networks.

**Structure-only tasks** In tasks such as traffic prediction and network flow classification, MPNNs can make predictions using purely structural information. We show that when nodes have uniform features, the prediction mechanism shifts—node degree becomes the effective feature, and a (k+1)-layer MPNN with uniform features predicts equivalently to a $k$-layer MPNN using node degree as the sole feature.

We also prove that combining featureless inputs with row normalization leads to degenerate predictions where MPNNs learn nothing.

In sum, structure-only tasks can be considered as a special type of Structure-feature Hybrid Tasks, where the node degree act as node feature.

## 5. Conclusions

This work demystifies Message Passing Neural Networks by revealing their computational essence: the message passing process is fundamentally a memory-efficient implementation of matrix multiplication operations. We establish that $k$-layer MPNNs aggregate information from $k$-hop neighborhoods through iterative computation, making them practical for large graphs where direct computation of powered adjacency matrices would be prohibitively expensive.

Through careful analysis of loop structures, we theoretically characterize how different types of loops influence $k$-hop neighborhood density. We demonstrate that common GNN practices, such as adding self-loops and converting directed graphs to undirected ones by adding reverse edges, significantly increase $k$-hop neighborhood density, potentially leading to over-smoothing.

Our analysis challenges two common misconceptions in the field: (1) performance degradation in deeper GNNs is not necessarily due to over-smoothing. For sparse directed graphs, deeper architectures are less susceptible to over-smoothing due to low connection density, yet their performance degrades due to vanishing gradients and overfitting from accumulated weights; and (2) deeper GNN architectures do not necessarily lead to over-smoothing as long as loop structures don't create dense $k$-hop connectivity.

Furthermore, we explained how GNNs work in structure-feature hybrid tasks and how for structure-only tasks, the node degree becomes the actual feature.

These insights offer theoretical understanding of how GNNs work and provides practical guidance for GNN architecture design, particularly regarding the choice of directed versus undirected aggregation, whether to add self-loops, and the selection of normalization strategies.

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
