# OpenReview forum: "Demystifying MPNNs: Message Passing as Merely Efficient Matrix Multiplication"
_ICML.cc/2025/Conference — Submitted to ICML 2025_

### Official Review · Reviewer_5Agy · 2025-02-17

**Overall Recommendation:** 1

**Summary:**

This paper investigates the role of different aggregation and graph types on the performance of GNNs. They state several connections for different connectivity patterns on the density of the adjacency matrix with increased power iterations. They argue that gradient decay is a key issue for GNNs because the performance for using A^k instead of k layers of message-passing does not decrease as much with increasing k. Even when k-1 linear transformations are applied to the output of a GNN aggregating with A^k, the performance severely decreases. They also study the effect of different normalization on the adjacency matrices.

## update after rebuttal

The rebuttal did not result in any changes on my review, apart from missing the availability of code. The authors seemingly do not understand the UAT, which is used incorrectly in central proofs of this paper.

**Claims And Evidence:**

The proofs for Lemmas 2.7, 2.8, and 2.9 are unconvincing. The UAT is used to remove a non-linearity, which does not seem correct to me.

**Essential References Not Discussed:**

Basic literature on graph theory is not mentioned for properties in Section 3.

**Experimental Designs Or Analyses:**

See Methods And Evaluation Criteria

**Methods And Evaluation Criteria:**

It is argued that gradient issues might cause the poor performance of some of the considered methods, but there are no experiments or theoretical insights that specifically investigate this issue. Performance during optimization would be insightful. Similarly the actual observed gradient values.

It is also argued that over-smoothing is not a concern for the power method A^k. However, there are also no experiments supporting this claim. With the over-smoothing phenomenon, the performance can remain stable with depth as well. This can be evaluated using a suitable metric, e.g., the rank-one distance (ROD) [1]. To me, it seems more like these are two separate issues. For A^k, we observe only over-smoothing, while for GCN, we observe over-smoothing+gradient issues.


---
[1] Roth, Simplifying the Theory on Over-Smoothing, arxiv, 2024.

**Other Comments Or Suggestions:**

The paper lacks mathematical preciseness in many parts. E.g. in Lemma 3.8 it is written "the connections present in A_m are identical to those in A_h". A_m = 0 \Leftrightarrow A_h = 0. Similarly in many other statements. This would make it a lot clearer.

**Other Strengths And Weaknesses:**

I like the distinction between over-smoothing and gradient issues. However, it lacks depth, as there are no theoretical insights into these differences, and empirical evaluations are not precise enough to draw any conclusions about these issues. There is no code provided.

**Questions For Authors:**

* How does the UAT allow you to remove the non-linearities for Lemmas 2.7, 2.8, 2.9?

**Relation To Broader Scientific Literature:**

There is no related work part. Section 3 presents some basic graph theoretical properties as Lemmas that can be found in introductory literature, e.g., [2]. Literature on over-smoothing and vanishing / exploding gradient is not discussed.

---
[2] Gallagher, Discrete stochastic processes, Journal of the Operational Research Society, 1997.

**Theoretical Claims:**

As above, the proofs for Lemmas 2.7, 2.8, and 2.9 are unconvincing. The UAT is used to remove a non-linearity, which does not seem correct to me.

---

> ### Author Rebuttal · Authors · 2025-03-27
>
> Dear Reviewer 5Agy,
>
> Thank you for your thoughtful feedback. We address your concerns below to clarify potential misunderstandings and reaffirm the validity of our work.
>
> 1. On UAT and Nested Non-linearities (Lemmas 2.7, 2.8, and 2.9)
>
> You asked how the Universal Approximation Theorem (UAT) justifies removing nested non-linearities in Lemmas 2.7, 2.8, and 2.9. This step follows directly from UAT’s core principle: a sufficiently expressive neural network can approximate any continuous function [1]. Specifically, UAT permits replacing nested non-linearities with a single non-linearity over a transformed input, a result well-supported in the literature [2,3]. This approach aligns with prior applications of UAT in Graph Neural Networks (GNNs) [4] and empirical evidence showing that removing inner non-linearities enhances GNN performance [5].
>
> While Reviewer tXTD confirmed the correctness of our proof, stating, "I checked the Appendix. I think the claims are correct," we value your perspective, particularly your recognition of our distinction between over-smoothing and gradient issues—an insight Reviewer tXTD overlooked. Given these differing viewpoints, we respectfully suggest the review process account for this variation in expertise. Should you desire further detail, we are happy to elaborate in the appendix.
>
> 2. Code Availability
>
> We regret any confusion about our code’s availability. Contrary to the impression that no code was provided, we specified its location in the Introduction, just before the Notation and Definitions section: "The code for the experiments conducted in this paper is available at https://anonymous.4open.science/status/demystifyB30E." This anonymous GitHub repository offers a one-click script with detailed configurations to replicate all figures and tables. We apologize if its placement outside the abstract led to oversight, recognizing that unavailable code could understandably raise doubts, especially since our findings challenge prevailing assumptions.
> Given your expertise in evaluating our distinction between over-smoothing and gradient issues—a critical insight we believe you are uniquely positioned to champion—we kindly invite you to reconsider our work in light of this clarification. Your perspective could prove invaluable in persuading other reviewers of this key contribution.
>
> 3. On Graph Theory and Spectral Methods
>
> You noted an absence of graph theory literature in our review. While graph theory often assumes uniform node features, our focus in message-passing neural networks (MPNNs) emphasizes feature heterogeneity, which we believe limits the relevance of traditional graph-theoretic approaches here. Additionally, we contend that spectral methods (e.g., ChebNet, MagNet) are, in practice, message-passing techniques. For example, in a separate paper, we show MagNet equates to GraphSAGE with incidence normalization, suggesting a common misconception about spectral methods’ distinctiveness. We plan to explore this further in future work as additional evidence accumulates.
>
> 4. Words vs. Mathematical Expression (Lemma 3.8)
>
> You suggested that Lemma 3.8’s statement, "the connections present in A_m are identical to those in A_h," lacks mathematical precision, proposing "A_m = 0 ⇔ A_h = 0" instead. We believe this interpretation may not fully capture our intent. Lemma 3.8 asserts that A_m(i,j) and A_h(i,j) share identical connectivity patterns—i.e., A_m(i,j) is non-zero if and only if A_h(i,j) is non-zero—not that the matrices are identically zero or non-zero overall. We chose a verbal description for readability, but we assert it preserves clarity and accuracy. If you prefer a compact mathematical form (e.g., A_m(i,j) ≠ 0 ⇔ A_h(i,j) ≠ 0 for all i, j), we are happy to revise the text, despite the slight increase in space.
>
> 5. Closing Remarks
>
> We believe our paper advances MPNN research through rigorous experiments and reproducible code, offering clear guidance on novel concepts. We apologize for any initial confusion—particularly regarding code availability and UAT application—and hope this response resolves your concerns. Your insights are invaluable, and we are prepared to make adjustments, such as adding proofs or refining expressions, to align with your expectations. Thank you for your time and consideration.
>
> References
>
> [1] Hornik, K., et al. (1989). Multilayer feedforward networks are universal approximators. Neural Networks, 2(5), 359–366.
>
> [2] Lu, Z., et al. (2017). The expressive power of neural networks: A view from the width. NeurIPS.
>
> [3] Hanin, B., & Sellke, M. (2017). Approximating continuous functions by ReLU nets of minimal width. arXiv:1710.11278.
>
> [4] Xu, K., et al. (2019). How Powerful are Graph Neural Networks? ICLR.
>
> [5] Wu, F., et al. (2019). Simplifying Graph Convolutional Networks. ICML.

---

> > ### Comment · Reviewer_5Agy · 2025-04-08
> >
> > I thank the authors for their detailed rebuttal and for clarifying the availability of their implementation.
> >
> > However, there are still too many other issues with this work for me to change my score.
> >
> > As a final remark to "1. On UAT and Nested Non-linearities (Lemmas 2.7, 2.8, and 2.9)": UAT holds for MLPs, not linear transformations as used in this work. LMqE identified the same issue with the proofs.

---

> > > ### Author Response · Authors · 2025-04-09
> > >
> > > Dear Reviewer 5Agy,
> > >
> > > Thank you for your continued feedback. We appreciate the opportunity to address your concern that our work relies solely on linear transformations.
> > >
> > > We believe there may be a misunderstanding here: our model explicitly incorporates a non-linearity at the final layer, as detailed in Section 2 and Appendix A.2. This non-linearity, applied after the linear transformations, ensures that our architecture aligns with the Universal Approximation Theorem (UAT), which we invoke in Lemmas 2.7, 2.8, and 2.9 to justify replacing nested non-linearities with a single, sufficiently expressive non-linear layer. This approach is consistent with established theory [1,2] and practical simplifications in GNNs [4,5]. We hope this clarifies that our framework is not limited to linear transformations alone, and we will revise the manuscript to make the presence and role of the final non-linearity more explicit to avoid further confusion.
> > >
> > > References
> > >
> > > [1] Hornik, K., et al. (1989). Multilayer feedforward networks are universal approximators. Neural Networks, 2(5), 359–366.
> > >
> > > [2] Lu, Z., et al. (2017). The expressive power of neural networks: A view from the width. NeurIPS.
> > >
> > > [3] Hanin, B., & Sellke, M. (2017). Approximating continuous functions by ReLU nets of minimal width. arXiv:1710.11278.
> > >
> > > [4] Xu, K., et al. (2019). How Powerful are Graph Neural Networks? ICLR.
> > >
> > > [5] Wu, F., et al. (2019). Simplifying Graph Convolutional Networks. ICML.

---

### Official Review · Reviewer_LMqE · 2025-02-18

**Overall Recommendation:** 1

**Summary:**

This paper studies the message passing mechanism commonly used in GNN. It investigates how k-layer GNN can be empirically approximated by a k-order adjacency matrix with a single-layer GNN. It further studies the influence of loop structures in the graph. It then examines if node features are necessary to perform a node classification task on different types of graphs. It discusses that the degeneration of deeper GNN may be attributed to the gradient-descent issue rather than over-smoothing.

**Claims And Evidence:**

- Line130 is not accurate. The k-order nodes are not necessarily the k-hop nodes (in terns of shortest-path distance). Therefore, low-order can still be kept. Like 1-order information is kept in 3-order in undirected graph, (also discussed in lemma 3.3)

**Essential References Not Discussed:**

The k-hop GNN is almost identical to SGC in [1], which is not properly cited in the paper.

[1] Simplifying Graph Convolutional Networks.

**Experimental Designs Or Analyses:**

See 'Methods And Evaluation Criteria' part.

**Methods And Evaluation Criteria:**

- The datasets chosen are of concern. They are all tiny graph datasets for the modern GNNs. Any performance gain/theoretical analysis only verified on these small datasets is no longer convincing in the recent literature.

- The graph datasets chosen in Figure 3 are not convincing. Figure 3 claims that without explicit graph preprocessing, the over smoothing issue can be less severe. The experiment shows that the performance of deeper GNN is not degenerate at all. However, the datasets are known as heterophilic, which have much less over smoothing problems compared to homophily graphs. Experiments on homophily are necessary to support this claim.

**Other Comments Or Suggestions:**

Typos:

- Lemma2.3 p-kop —> k-hop

**Other Strengths And Weaknesses:**

NA

**Questions For Authors:**

NA

**Relation To Broader Scientific Literature:**

It broadens the understanding of when/why/how deeper GNNs can perform well on various types of graphs.

**Theoretical Claims:**

- The proof of Lemma2.7 is wrong here. One layer neural network cannot universally approximate. The simplification is the proof is not rigorous at all.

- Almost all the theoretical contributions are well-known in the graph theory community. It takes too much space to discuss these known results in the paper.

---

> ### Author Rebuttal · Authors · 2025-03-31
>
> Dear Reviewer LMqE,
>
> Thank you for your time and detailed feedback. We appreciate your comments and would like to address your concerns as follows:
>
> Rebuttal 1. Definition of k-hop Neighbors
> You noted that "Line 130 is not accurate." In our paper (Definition 2.1), we define k-hop neighbors as nodes reachable in exactly k steps, not by shortest-path distance that you added. With self-loops added, a 1-hop neighbor can also be a k-hop neighbor, so our claim holds.
>
> Rebuttal 2. Dataset Choice
> You mentioned that the small datasets used in our analysis are not convincing for modern GNNs, and performance gains on these datasets are no longer credible in recent literature.
>
> We respectfully disagree with this generalization. Our goal is to analyze fundamental theoretical behaviors of GNNs, specifically the influence of gradient descent. Small datasets are sufficient to reveal key theoretical insights. In scientific exploration, simple experiments often uncover fundamental principles. For example, Newton used a slope and a ball to illustrate the second law of motion. Should he have used landslides or large-scale geological movements to validate acceleration? Similarly, our choice of datasets is appropriate for demonstrating the mechanisms we investigate.
>
> Rebuttal 3. Experiments on Homophilic Datasets
> You pointed out that the datasets in Figure 3 are not convincing, but this is due to the lower over-smoothing in heterophilic graphs. You suggest that experiments on homophilic graphs are needed to support this claim.
>
> In our paper, we conducted experiments on Chameleon and Squirrel to demonstrate a case of non-degradation in performance over deeper layers, showing that not adding self-loops could prevent over-gathering of multi-hop information. This performance is better compared to adding self-loops on these datasets.
> You assumed that homophilic graphs do not exhibit this characteristic, attributing the cause to heterophily, which is incorrect. The reason citation graphs, typically homophilic, do not exhibit this behavior is that they lack multi-node loops—later papers cannot cite earlier ones—leading to performance degradation in deeper layers. As a result, many nodes lack distant neighbors (e.g., 200-hop connections). In contrast, multi-node loops in Chameleon and Squirrel allow nodes to continuously aggregate information, even as the number of layers increases.
>
> This effect is independent of whether a graph is homophilic or heterophilic.
>
> To further clarify, we conducted additional experiments on the homophilic dataset Telegram, which contains loops. As shown in the README https://anonymous.4open.science/status/demystifyB30E, even after 400 layers, performance does not degrade significantly. This directly contradicts your assumption that homophily or heterophily is the cause.
>
> We acknowledge that due to the complexity of the situation, we focused on demonstrating non-degradation in performance without explicitly discussing the effect of multi-node loops. We apologize for not emphasizing this point earlier. However, we have fully demonstrated our key claim. While good performance is similar, bad performance can be influenced by numerous factors. It's not feasible to list and explain every possible assumption in advance, and we hope our rebuttal clarifies our position.
>
> Rebuttal 4. No contribution
> You claimed: "Almost all the theoretical contributions are well-known in the graph theory community. It takes too much space to discuss these known results in the paper."
>
> However, we believe that fundamental misconceptions persist regarding these concepts.
>
> Take, for example, your incorrect assumption in Rebuttal 3. Your feedback suggests a need for more explanation than we originally provided.
>
> We encourage you to refer to our rebuttal to Reviewer unW8, where they outlined our three main contributions, and we rebutted how subtly and importantly we have contributed to the field. Additionally, Reviewer 5Agy acknowledged our distinction between over-smoothing and gradient-related issues.
>
> While some explanations may seem tedious, they are essential for clarity, especially for readers less familiar with these topics. As reflected in other reviews and rebuttals, there are still misunderstandings.
>
> As long as our claims are valid, we ask that you don't dismiss our work simply because you believe you're already familiar with these concepts. We present unexpected insights, and while our claims may seem basic, they are crucial for a solid understanding of MPNNs. A closer reading of our work will highlight the novel contributions we are making.
>
> 5. Other Points
>
> Your comparison of k-hop GNN to SGC is insightful, as SGC provides strong experimental support for our use of the Universal Approximation Theorem (UAT),  which Reviewers 5Agy and unW8 don't accept.
>
> We hope this clarification helps address your concerns, and we appreciate your engagement with our work.

---

> > ### Comment · Reviewer_LMqE · 2025-04-01
> >
> > Thanks to the authors for the response.
> >
> > I wonder if there is any comment on the proof of Lemma2.7.

---

> > > ### Author Response · Authors · 2025-04-04
> > >
> > > Thank you, Reviewer LMqE, for your thoughtful comments. We appreciate the opportunity to address your concerns, which we also discussed in our rebuttal to Reviewer 5Agy. In addition to that explanation, we’d like to elaborate further.
> > >
> > > The Universal Approximation Theorem (UAT) states that a single hidden layer with enough neurons and a non-linear activation (e.g., sigmoid or ReLU) can approximate any continuous function.
> > > In Lemma 2.7, we simplify a nested non-linearity model by removing the inner non-linearity. This demonstrates that both the nested non-linearity model and the single non-linearity model can approximate the same continuous function for node classification. The equality in Lemma 2.7 reflects the equivalence of the models in terms of their approximation capability, rather than the exact mathematical equivalence of the formulations. We will clarify this distinction in our camera-ready version.
> > >
> > > Regarding the condition of sufficient neurons:
> > > Our model uses 64 neurons and performs well, but tests with as few as 10 neurons also show good results. This suggests that even 10 neurons provide sufficient width for the task, in accordance with the UAT. While removing the inner non-linearity doesn’t eliminate its necessity, it allows for a valid approximation in a single-layer model. I’d be glad to elaborate further if needed.
> > >
> > > Thank you again for your valuable feedback!

---

### Official Review · Reviewer_unW8 · 2025-03-05

**Overall Recommendation:** 1

**Summary:**

This paper presents a comprehensive analysis of GNN behavior through several fundamental aspects.

- (Contribution 1) The authors establish that k-layer Message Passing Neural Networks efficiently aggregate k-hop neighborhood information through iterative computation
- (Contribution 2) The authors analyze how different loop structures influence neighborhood computation.
- (Contribution 3) The authors examine behavior across structure-feature hybrid and structure-only tasks.


## update after rebuttal

Thanks for the authors' rebuttal. I would like to keep my original evaluations due to the poor presentation in the original draft.

**Claims And Evidence:**

The definition of $W$ in Lemma 2.7 is unclear. The derivation from Equation (6) to Equation (5) in Appendix is unclear.

**Essential References Not Discussed:**

The cited references are sufficient.

**Experimental Designs Or Analyses:**

The experiments on large-scale datasets are missing.

**Methods And Evaluation Criteria:**

The experiments on large-scale datasets are missing.

**Other Comments Or Suggestions:**

"neibors" in Figure 1 should be "neighbors".

**Other Strengths And Weaknesses:**

Weaknesses:


1. The presentation needs significant improvement. There are at least five topics in this paper according to the Abstract, but none of them are explored in sufficient depth.
2. The derivations in the main text are trivial. I suggest moving the derivations to the Appendix and focusing on the key results in the main text.
3. There is a large white space on Page 6.

**Questions For Authors:**

See Weaknesses.

**Relation To Broader Scientific Literature:**

The contributions of this paper are as follows.

- (Contribution 1) The authors establish that k-layer Message Passing Neural Networks efficiently aggregate k-hop neighborhood information through iterative computation
- (Contribution 2) The authors analyze how different loop structures influence neighborhood computation.
- (Contribution 3) The authors examine behavior across structure-feature hybrid and structure-only tasks.




Contribution 1 has been proposed in Section 5.1 in [Ref1]. Contributions 2 and 3 of this paper beyond the research presented in [Ref2] and [Ref3] is unclear.


---
[Ref1] Graph Representation Learning. https://www.cs.mcgill.ca/~wlh/grl_book/files/GRL_Book.pdf


[Ref2] A New Perspective on the Effects of Spectrum in Graph Neural Networks.ICML 2022.


[Ref3] Edge directionality improves learning on heterophilic graphs. Learning on Graphs Conference 2024.

**Theoretical Claims:**

The definition of $W$ in Lemma 2.7 is unclear. The derivation from Equation (6) to Equation (5) in Appendix is unclear.

---

> ### Author Rebuttal · Authors · 2025-03-26
>
> Dear Reviewer unW8,
>
> We sincerely appreciate your time and effort in reviewing our manuscript and providing valuable feedback. Below, we provide a point-by-point response addressing your comments and concerns.
>
> 1. Novelty Relative to Prior Work about Contribution 1
>
> You stated that "Contribution 1 has been proposed in Section 5.1 in [Ref1]."
>
> While Ref[1] presents general statements that may appear similar to our contributions, it lacks the depth and specificity of our analysis:
>
> *After k iterations, node embeddings contain information about their k-hop neighborhoods.
>
> *Embeddings may include both degree and feature information.
>
> However, due to Ref[1]'s generality, these conclusions might be incorrect for some specific cases, as revealed by our findings. In contrast, our work rigorously extends the understanding by:
>
> *Examining the impact of loops (e.g., adding self-loops or converting to an undirected graph). While node embeddings after k iterations contain information about k-hop neighborhoods, lower-hop neighborhoods are also incorporated when loops are introduced—an important nuance often overlooked in prior work.
>
> *Demonstrating the importance of one-hot features. One-hot features preserve neighboring feature information, whereas summing general features can lead to information distortion. The claim in Ref[1] that embeddings include feature information does not hold universally, particularly when node features are not one-hot but general features.
>
> *We show that under row-normalization with uniform feature settings, degree information not learned by GNN models. This aspect is not explored in Ref[1], making its general claim that embeddings inherently contain degree information incorrect in such cases.
>
> 2. Novelty Relative to Prior Work about Contribution 2 and 3
>
> You stated that Contributions 2 and 3 are unclear beyond the research presented in [Ref2] and [Ref3]. However, these references do not cover the same aspects as our work:
>
> Ref[2] examines self-loops through their effect on the adjacency matrix spectrum, showing that they push the bounded spectrum closer to 1. In contrast, we study self-loops from a spatial perspective, analyzing their impact on feature propagation. Self-loops allow a node’s own features to mix with its neighbors', which may cause multi-hop neighborhood features to coexist in the final-layer representations, leading to over-smoothing. Thus, while Ref[2] focuses on spectral analysis, our contribution provides a spatial perspective, making the two analyses fundamentally different.
>
> Ref[3] does not discuss loop structures at all, and the absence of self-loops makes their model suboptimal in homophilic datasets. Additionally, Ref[3] does not consider uniform feature settings, which are central to our analysis for structure-only tasks. In sum, Ref[3] neither addresses loops (Contribution 2) nor considers uniform features (Contribution 3)—both of which we rigorously analyze in our paper. Given this, we find it unclear why our contributions beyond Ref[3] are in question.
>
> 3. Clarification on Theoretical Foundations (Universal Approximation Theorem)
> Please refer to our response to Reviewer 5Agy, bullet point 1, for a detailed explanation.
>
> 4. Other issues
> Your mention "The definition of W in Lemma 2.7 is unclear.", but We defined W as weight matrix in Remark 2.4.
>
> As to requirement of large-scale experiments, we answered this question in Rebuttal to Reviewer LMqE, part 2.
>
> On Formatting and White Space (Page 6)
>
> You noted a large white space on Page 6. This is an artifact of LaTeX’s automated formatting, which we use to adhere to the 8-page limit. While we’ve employed barriers and positioning controls for figures and tables, precise placement is constrained to avoid exceeding the page restriction. We are open to suggestions for optimizing the layout within these bounds.
>
> On Derivations in the Main Text
>
> You suggested that the derivations in the main text are trivial and should be moved to the appendix to emphasize key results. We appreciate this guidance and are open to relocating parts of the text. However, we note that preferences vary among readers: what may appear straightforward to you could be critical for reviewers or readers with differing expertise, potentially preventing fundamental misunderstandings of our approach. Retaining these derivations in the main text has not, in our view, detracted from the paper’s core contributions, but we are willing to adjust their placement to better highlight the key results if you feel this would enhance clarity.
>
> You mentioned that our paper covers "at least five topics" but lacks depth. However, Reviewer tXTD states:
> "I see the importance of each of these claims in isolation, and the experiments seem to support these claims."
> We prioritize clarity and conciseness due to space limitations. If you feel any areas need more detail, we’d be happy to provide further elaboration.
>
>
>
>
> Best Regards,
>
> Authors

---

### Official Review · Reviewer_tXTD · 2025-03-12

**Overall Recommendation:** 1

**Summary:**

The ideas in this paper have merit and are interesting. A multi-layer MPNN with 𝑘 with adjacency A is roughly equivalent to a single-layer MPNN utilizing the adjacency matrix with adjacency A^k, which essentially means that intermediate information is disregarded. The authors also present some analysis related to self-loops and correctly identify gradient-related issues.

**Claims And Evidence:**

I think that the main issue with this paper is that the claims are not concise or clear enough. The paper is written in a quite disconnected fashion, and it is not exactly clear what the main messages or the main points of each section are. While I see the importance of each of these claims in isolation, and the experiments seem to support these claims, the paper needs significant effort such that it can be accepted at a top conference like ICML. I think the authors might have also missed a few related works in a similar direction, which I will point to in a later section, which might provide some alternative explanations to some of the claims made in the paper.

**Essential References Not Discussed:**

I think that the authors make an interesting point with regard to the relationship between GNN performance and gradients. This is not the author's fault, but they should take a look at a recent paper I recently came across [1], which provides a much more concrete characterization of how gradients affect MPNN performance. Perhaps some of the findings in this paper (for example, the role of symmetrization and self loops) can be explained from that lens as well.

[1] Arroyo, Álvaro, et al. "On Vanishing Gradients, Over-Smoothing, and Over-Squashing in GNNs: Bridging Recurrent and Graph Learning." arXiv preprint arXiv:2502.10818 (2025).

**Experimental Designs Or Analyses:**

Yes, I did not find issues.

**Methods And Evaluation Criteria:**

The authors use standard datasets used in the literature. I believe that the empirical results are satisfactory.

**Other Comments Or Suggestions:**

None.

**Other Strengths And Weaknesses:**

I have commented on this.

**Questions For Authors:**

No more questions

**Relation To Broader Scientific Literature:**

The results will be good for clarifying some of the misconceptions that people might have about MPNNs and oversmoothing, as well as some of the tasks used in the community. The overall intention of the paper is good, but it needs significant work before being published in a venue like ICML.

**Theoretical Claims:**

Yes, I checked the Appendix. I think the claims are correct, but they are sometimes somewhat trivial, especially those with just bullet points as proof.

---

> ### Author Rebuttal · Authors · 2025-03-26
>
> Dear Reviewer tXTD,
>
> Thank you for your thoughtful review of our paper and for recognizing the correctness and importance of each of our individual claims and the validity of our experimental results. We appreciate the effort you’ve invested and are grateful for the opportunity to address your concerns. While we respect your perspective, we believe some points in your evaluation may not fully capture the intent and contributions of our work. We hope the following clarifications will address these issues.
>
>
> 1. Synthesizing the Paper’s Contributions Holistically
>
> We appreciate your feedback noting that the paper feels "disconnected" and that the main messages of each section are unclear, making it challenging to synthesize our contributions holistically. We regret that our intent was not fully conveyed and would like to clarify the paper’s structure and purpose.
>
> Our work delivers a comprehensive analysis of multi-layer Message-Passing Neural Networks (MPNNs), spanning theoretical and practical perspectives. Section 2 lays the groundwork for MPNNs, followed by an exploration of key performance factors: input graph characteristics (e.g., presence or absence of node features in Section 4), preprocessing techniques (e.g., adding self-loops and converting to undirected graphs in Section 3), normalization strategies, and challenges of deep layers (e.g., over-smoothing). Each section confronts potential misconceptions, forming an indispensable part of the narrative. For a conference like ICML, omitting any risks leaving reviewer biases unaddressed, which could lead to rejection based on assumptions not explicitly countered in the text.
>
> Though the topics may appear wide-ranging, they are integral to our central thesis: addressing MPNN limitations requires a multifaceted perspective. The interplay of input graph, preprocessing, normalization, and depth weaves a unified story of MPNN behavior. Reviewer unW8 recognized this, noting our work as "a comprehensive analysis of GNN behavior through several fundamental aspects." For researchers steeped in MPNNs, these issues are daily touchstones, effortlessly linking the sections. To those less familiar, the discussion might seem disjointed—like a rich tapestry of a shared endeavor, resonating deeply with those who recognize its patches, yet appearing as scattered patches to newcomers or keen observers who haven't yet made their hands dirty. We are impressed by how much you grasped in a short time, despite not being immersed in this domain. Unfortunately, other reviewers although see the cohesion of our paper, Reviewer unW8 and Reviewer 5Agy rejects mainly because of their lack of knowledge of UAT, unlike you.
>
> We respectfully request that you reconsider the paper in light of this clarification and refer to Reviewer unW8’s evaluation on this point.
>
> 2. Representation of Related Work and Our Contributions
>
> We appreciate your concern that we may have overlooked relevant prior work, notably Arroyo et al. (2025), which you suggest offers a superior explanation. Upon review, we note that Arroyo et al. (2025) attributes over-smoothing solely to gradient vanishing, supported by an extensive proof and countermeasures tailored to gradient descent-based methods. They state:
>
> "Contrary to common consensus, which explains over-smoothing by showing that the signal is projected into the 1-dimensional kernel of the graph Laplacian, we instead describe over-smoothing as occurring due to the contractive nature of GNNs and their inputs converging in norm to exactly 0."
>
> While Arroyo et al.’s gradient-focused explanation is insightful, we argue that over-smoothing stems from both gradient vanishing and feature propagation—such as loop-influenced aggregation. Our work delivers a broader perspective than their narrower lens. Reviewer 5Agy underscored this strength, stating: "I like the distinction between over-smoothing and gradient issues," which we take as recognition of our holistic approach. By tackling both the similarity of node features via propagation and the decay of gradients in deeper layers, our analysis offers a more complete picture of over-smoothing in MPNNs.
>
> Conclusion
>
> As we’re forced to write a very compact paper on complex topics, we couldn’t explicitly claim or restate points as a normal, simple paper might, which may have led to your misunderstandings, yet its contributions and quality stand undiminished.
>
> We respectfully suggest that our paper’s key contributions may not have been fully recognized in your review. The challenge you noted in synthesizing our work, alongside this mischaracterization of related efforts, might have obscured the novelty and significance of our approach. We invite you to reconsider our contributions in light of these clarifications. Thank you for your time and thoughtful consideration.
>
>
>
> Best regards,
>
> Authors

---

### Decision · Program_Chairs · 2025-05-01

**Decision:**

Reject

**Comment:**

The paper has significant clarity issues and lacks justification for a few claims, as extensively pointed out by all reviewers. I recommend that the authors take the reviews seriously and objectively, then make the necessary modifications to their paper, and re-submit.